# Assessment of the Salivary Concentrations of Selected Immunological Components in Adult Patients in the Late Period after Allogeneic Hematopoietic Stem Cell Transplantation—A Translational Study

**DOI:** 10.3390/ijms25031457

**Published:** 2024-01-25

**Authors:** Aniela Brodzikowska, Barbara Kochańska, Agnieszka Bogusławska-Kapała, Izabela Strużycka, Bartłomiej Górski, Andrzej Miskiewicz

**Affiliations:** 1Department of Conservative Dentistry, Medical University of Warsaw, 02-091 Warsaw, Poland; 2Department of Conservative Dentistry, Medical University of Gdańsk, 80-210 Gdańsk, Poland; barbara.kochanska@gumed.edu.pl; 3Department of Comprehensive Dental Care, Medical University of Warsaw, 02-091 Warsaw, Poland; aboguslawska@wum.edu.pl (A.B.-K.); istruzycka@wum.edu.pl (I.S.); 4Department of Periodontology and Oral Diseases, Medical University of Warsaw, 02-091 Warsaw, Poland; bgorski@wum.edu.pl (B.G.); andrzej.miskiewicz@wum.edu.pl (A.M.)

**Keywords:** alloHSCT recipients, cGvHD, lactoferrin, lysozyme, sIgA, whole resting saliva, whole stimulated saliva, oral complications

## Abstract

(1) The aim of the study was to analyze the salivary concentrations of lysozyme, lactoferrin, and sIgA antibodies in adult patients in the late period after allogeneic stem cell transplantation (alloHSCT). The relationship between these concentrations and the salivary secretion rate and the time elapsed after alloHSCT was investigated. The relationship between the concentrations of lysozyme, lactoferrin, and sIgA and the titer of the cariogenic bacteria *S. mutans* and *L. acidophilus* was assessed. (2) The study included 54 individuals, aged 19 to 67 (SD = 40.06 ± 11.82; Me = 39.5), who were 3 to 96 months after alloHSCT. The concentrations of lysozyme, lactoferrin, and sIgA were assessed in mixed whole resting saliva (WRS) and mixed whole stimulated saliva (WSS). (3) The majority of patients had very low or low concentrations of the studied salivary components (WRS—lysozyme: 52, lactoferrin: 36, sIgA: 49 patients; WSS—lysozyme: 51, lactoferrin: 25, sIgA: 51 patients). The levels of lactoferrin in both WRS and WSS were statistically significantly higher in the alloHSCT group than in the control group (CG) (alloHSCT patients—WRS: M = 40.18 μg/mL; WSS: M = 27.33 μg/mL; CG—WRS: M = 17.58 μg/mL; WSS: 10.69 μg/mL). No statistically significant correlations were observed between lysozyme, lactoferrin, and sIgA concentrations and the time after alloHSCT. In the group of patients after alloHSCT a negative correlation was found between the resting salivary flow rate and the concentration of lactoferrin and sIgA. The stimulated salivary flow rate correlated negatively with lactoferrin and sIgA concentrations. Additionally, the number of *S. mutans* colonies correlated positively with the concentration of lysozyme and sIgA. (4) The concentrations of non-specific and specific immunological factors in the saliva of patients after alloHSCT may differ when compared to healthy adults; however, the abovementioned differences did not change with the time after transplantation.

## 1. Introduction

There has been a significant increase in interest in using the body’s natural defense factors for fighting and preventing local and systemic infections [1]. This line of research may be of particular importance for patients with prolonged impaired immunity who are prone to both local and systemic infections. This group includes, among others, patients in the late period (>100 days) after allogeneic hematopoietic stem cell transplantation (alloHSCT). The regeneration of the hematopoietic system after this medical procedure is a long-term process [2]. Qualitative and quantitative disorders of cellular and humoral immunity in many patients persist even several years after transplantation [3]. This results in the frequent occurrence of various types of complications, which often affect the oral cavity [4]. The most serious of these include infections, inflammation of the oral mucosa, and decreased salivation [5,6,7]. As a result of lowered salivary secretion, its composition may change [8], including changes in the concentrations of non-specific and specific immunity factors [7,9,10,11,12]. Altered salivary composition often lead to perturbation of the function and composition of the oral microbiome, causing dysbiosis with an associated risk of complications, such as rampant dental caries or periodontal disease [13]. Hyposalivation is also a risk factor for secondary infection, i.e., mucositis with ulcers [14]. The disturbance of the oral environment balance poses a particular threat to immunodeficient patients in the late period after alloHSCT. A lot of additional components may contribute to the rapid development of various oral diseases in this group [4,5]. Among which there are the following: high sugar consumption (frequent in the case of taste disorders) [15], low frequency of toothbrushing (patients with depression) [16], occasional or absence of tooth hard tissue remineralization (difficulty moving and rare professional dental procedures) [17,18], inaccurate oral hygiene (limiting factors in oral selfcare and professional oral health care, e.g., microstomia, pain) [4,5]. 

Decreased levels of defensive salivary factors may promote oral mucosal damage or prolonged healing [19]. Some authors indicate an increased risk of cancer, i.e., lysozyme can directly activate immune cells, or it can liberate peptidoglycans and polyribopyrimidinic acids from bacteria responsible for immunopotentiation [19,20]. It is also suspected that the deficiency of some salivary biomarkers may contribute to the pathogenesis of mucosal cGvHD [21]. Understanding the mechanisms responsible for oral diseases in patients in the late period after alloHSCT is essential for proper individual treatment planning.

A review of the literature indicates insufficient knowledge about defensive salivary parameters in patients during the late period after alloHSCT [7]. The research concerned the early period after transplantation (<100 day) or a maximum period of up to one year after the procedure [7,22]. Therefore, it does not consider the influence of long-term general and local complications on the composition of saliva [3]. In addition, the salivary content of the immunity factors was often examined in patients after chemotherapy followed by autologic hematopoietic stem cell transplantation (autoHSCT)—a procedure with different specificity [23,24]. Also, several valuable examinations were conducted, but on small populations [21]. Some of the most important factors of non-specific immunity contained in saliva are lysozyme and lactoferrin [25,26,27,28,29,30]. Lysozyme is a cationic protein, the source of which in saliva is secretion by the serous cells of the salivary glands, gingival crevicular fluid, and the granules of neutrophils, monocytes, and macrophages [30]. This enzyme has a wide range of non-specific effects on pathogenic microorganisms. It is mainly active against the Gram-positive bacteria *S. mutans* (*Streptococcus mutans*) and *L. acidophilus* (*Lactobacillus acidophilus*), and to a lesser extent against Gram-negative bacteria and some fungal species [26,27,28,29,30,31]. Lysozyme breaks down microbial cells by hydrolyzing bonds between N-acetyl-glucosamine and N-acetylmuramic acid of peptidoglycan in the bacterial cell wall [32]. In addition, it has the ability to activate bacterial autolysins and the ability to inhibit glucose absorption and acid production by bacteria [26,27,28,29]. As observed, lysozyme can also cause the aggregation of microorganisms and inactivate some viruses [26]. Moreover, lysozyme has regenerative and anti-inflammatory roles in the human organism [33]. Studies have shown that the local treatment of oral mucositis caused by radiotherapy/chemotherapy with lysozyme-based compounds led to the healing of oral ulcerations and decreased pain [34].

Lactoferrin is a transferrin glycoprotein, produced by epithelial cells, commonly found in secretions such as milk, saliva, tears, and gastrointestinal secretions [35]. It is also one of the main components of granularity of neutrophils from which it is released during inflammatory reaction [36,37]. The main source of lactoferrin in saliva is epithelial cells of the salivary serous glands. Lactoferrin is biostatic or biocidal against many species of bacteria (including *Streptococcus mutans*, *mitis*, *salivarius*), fungi (*Candida albicans*) and viruses (HPV1—*Human Papillomavirus*, HIV—*Human Immunodeficiency Virus*) [38,39]. The antimicrobial action of lactoferrin is related to, inter alia, its high ability to bind iron ions, preventing the use of this element by pathogens [40]. Additionally, lactoferrin disintegrating under the influence of pepsin releases peptides with direct bacteriostatic and antifungal activity [29]. Lactoferrin also has an immunomodulating effect, e.g., by stimulating lymphocytes to increase production of TNF-α (tumor necrosis factor) and INF-γ (interferon), and also by stimulating neutrophils to phagocytosis and secreting interleukin Il-8. Moreover, lactoferrin regulates the production of GM-CSF (granulocyte-macrophage colony-stimulating factor) by macrophages and disrupts adherence of microorganisms to tissues, including *S. mutans* to enamel hydroxyapatite [41].

The dominant immunoglobulin in saliva is IgA, present there mainly in the secretory form—sIgA (secretory immunoglobulin A) [30]. sIgA is produced by plasma cells located around the outlet ducts of salivary glands, primarily parotid and submandibular glands [42]. The sources of IgA are also gingival crevicular fluid and plasma seeping through damaged mucosal barrier [42]. The secretory form of IgA co-forms the first line of defense against colonization of mucous membranes by pathogenic microorganisms and is one of the main mediators of saliva responsible for homeostasis of oral tissues. The basic task of sIgA is to limit contact of microorganisms with the mucosa surface [43]. The mechanism of action of immunoglobulin is mainly based on its binding with microbial adhesins, which in turn prevents them from connecting with the host’s epithelial cell receptors and from penetrating deep into the system. Moreover, sIgA may cause agglutination of bacteria [43,44]. It also has the ability to neutralize toxins and bacterial enzymes by blocking receptors that bind toxins to target cells. The range of biological influence of sIgA includes bacteria, fungi, and viruses [45]. An important feature of IgA is its ability to interact with non-specific oral defense factors. Lysozyme and sIgA have been found to collaborate with salivary mucins in immobilization of microorganisms [46].

The aim of the study was to analyze the salivary concentrations of lysozyme, lactoferrin, as well as sIgA antibodies in patients in the late period after alloHSCT. It was also assessed whether concentrations of lysozyme, lactoferrin, and sIgA in saliva changed significantly depending on the time elapsed after transplantation. In addition, the subject of the investigation was the relationships between the concentrations of lysozyme, lactoferrin, and sIgA in mixed saliva and the rate of its secretion, as well as the relationships between the concentrations of lysozyme, lactoferrin, and sIgA and the titer of cariogenic bacteria *S. mutans* and *L. acidophilus*. It was assumed that due to general immunodeficiency during the late period after alloHSCT [2,3] and possible local long-lasting hyposalivation [5,6,7], disturbances in salivary immune factors would occur. As mentioned, this knowledge may be important for the development of proper oral treatment/preventive plan for the individual patient.

## 2. Results

### 2.1. Concentration of Lysozyme, Lactoferrin, sIgA in Unstimulated and Stimulated Saliva in Patients after alloHSCT

The concentrations of lysozyme, lactoferrin, sIgA in the resting and stimulated saliva results in the study group and in the control group were compared. Mann–Whitney U tests were performed. According to the presented data, the majority of patients after alloHSCT had very low or low concentrations of the studied salivary components (Table 1; Figure 1 and Figure 2). As can be seen from the data presented in Table 1 two statistically significant differences were observed. The levels of lactoferrin in both study variants were higher in the alloHSCT group. The power of the recorded effects, measured by Pearson correlation coefficient (r), was high. There were no statistically significant differences in the remaining variables (Table 1). Statistically significant results are presented in Figure 3 and Figure 4.

### 2.2. Correlation between Lysozyme, Lactoferrin, sIgA Concentrations and the Time after alloHSCT

In the next step, the correlation between lysozyme, lactoferrin, sIgA concentrations and the time after alloHSCT was examined (Table 2). Spearman’s ρ rank correlation analysis was performed. As can be seen from the data presented in Table 2, no statistically significant correlations were observed. The studied variables were not significantly related to each other.

### 2.3. Correlation between Lysozyme, Lactoferrin, sIgA Concentrations, and the Unstimulated and Stimulated Saliva Flow Rate

The correlation between lysozyme, lactoferrin, sIgA concentrations, and the rate of unstimulated and stimulated salivation was examined. We used Spearman’s ρ rank correlation analysis, in parallel for the study group and the control group. In the alloHSCT patient group, unstimulated and stimulated saliva median flow rates were as follows: M = 0.25 mL/min (Me = 0.27; SD = 0.18) and M = 0.79 mL/min (Me = 0.60; SD = 0.51), respectively. In the control group unstimulated and stimulated saliva median flow rates were statistically significantly higher (*p* < 0.05): M = 0.43 mL/min (Me = 0.44; SD = 0.23) and M = 1.29 mL/min (Me = 1.22; SD = 0.51), respectively. Both in the group of patients after alloHSCT and in the control group, a negative correlation was found between the resting salivary flow rate and the concentrations of lactoferrin and sIgA (Table 3). The remaining correlations turned out to be statistically insignificant. On the other hand, the stimulated salivary flow rate correlated negatively with lactoferrin and sIgA concentrations. In the control group, no correlation turned out to be statistically significant (Table 3).

### 2.4. Correlation between Lysozyme, Lactoferrin, sIgA Concentrations and S. mutans and L. acidophilus Titers

In the next step, the correlation between lysozyme, lactoferrin, sIgA concentrations in stimulated saliva and the titer of cariogenic bacteria—*S. mutans* and *L. acidophilus*—was examined. Spearman’s ρ rank correlation analysis was performed in parallel for the study group and the control group. The results are presented in Table 4. In the group of patients after alloHSCT, two statistically significant correlations were observed: the number of *S. mutans* correlated positively with the concentration of lysozyme and with the concentration of sIgA in stimulated saliva. The power of these correlations was moderately high. In the control group, only one statistically significant correlation was noted. The *S. mutans* titer correlated positively with sIgA concentration in stimulated saliva. The power of this correlation was moderately high. The remaining correlations turned out to be statistically insignificant. Analogous relationships were examined for the *L. acidophilus* titer in stimulated saliva. The studied correlations were not statistically significant (Table 4).

## 3. Discussion

The results of our research indicate that concentrations of lysozyme in resting and stimulated saliva were not statistically higher in subjects after alloHSCT compared to the control group. It was noticeable, however, that in the study group of patients after transplantation, there were very diverse values of lysozyme concentration. For comparison, in the control group the fluctuations of the examined parameter were relatively small. The observed differentiations of the lysozyme values might be the result of factors such as those that follow: mucosal damage as a complication of oral cGvHD (an increase of the values); chronic stress (a decrease of the values) [47]; some drugs, e.g., antifungal ones (a decrease of the values) [48]; some accompanying disease, e.g., hypertension (an increase of the values) [48]. Unfortunately, no similar comparison was found in the literature for patients after alloHSCT [7]. The results concerning lysozyme concentrations in saliva of the patients during chemotherapy (without following alloHSCT) are contradictory. Some authors examining subjects undergoing chemotherapy and healthy individuals did not find significant differences in concentrations of lysozyme between these groups [49]. On the other hand, Sun et al. observed a significant increase in lysozyme concentrations in patients undergoing chemotherapy due to hematopoietic malignancies [20]. In our research, we did not observe a statistically significant relationship between the secretion rate of resting and stimulated saliva and the level of lysozyme. But Yeh et al., examining a population of 595 randomly selected individuals, in the aspect of oral mycosis, reported that the levels of lysozyme inversely depended on the rate of stimulated saliva secretion [26]. According to our observations, the levels of lysozyme in whole mixed saliva, both in resting and stimulated saliva, did not significantly depend on the time elapsed after alloHSCT. The above results could not be compared with observations of other authors, because available literature did not contain publications relating to the presented issues. The results presented in this study indicate that concentrations of lysozyme were statistically significantly higher in those subjects after alloHSCT who had a high titer of *S. mutans*. Other research proved lysozyme’s inhibitory effect against cariogenic bacteria [50]. As was demonstrated, this relationship can be used in practice, e.g., in professional and home dental care (dental paste or mouthwashes) [14].

The results of our research concerning lactoferrin levels in saliva are like those obtained by other authors [22]. It was found that the levels of lactoferrin in both resting and stimulated saliva were statistically significantly higher in the group of patients after alloHSCT compared to the control group. According to Keskin et al. it can indicate on an ongoing inflammatory process in the oral cavity [37]. The observed mean concentrations of lactoferrin in patients after transplantation remained at the level similar to those reported by Eliasson et al. in patients with Sjogren’s syndrome [51]. Also, Imanguli et al., examining saliva of patients who were 6 months after alloHSCT, found numerous changes in the protein profile compared to the period before transplantation, including, inter alia, significantly increased concentrations of lactoferrin [22]. The analysis of the results of studies presented in this paper showed that the time elapsed after alloHSCT did not have a decisive impact on lactoferrin concentrations, whereas they statistically significantly depended on the rate of saliva secretion, both resting and stimulated (it was a negative correlation). Sikorska noticed a similar dependence while examining the salivary levels of this protein in healthy subjects [40]. An analysis of literature and the results of our own research suggest that the reason for increased concentrations of lactoferrin in the studied group of patients after alloHSCT could be, among other things, gingival inflammation. As many as 56.8% of patients had symptoms of gingivitis of varying severity (the study is not included in the table). In addition, Almståhl, while examining patients with Sjögren’s syndrome, reported inflammation of marginal periodontium as a possible cause of an increase in the concentrations of lactoferrin in these patients’ saliva [11]. Such a dependence was also noted by Groennik et al. who examined saliva of generally healthy subjects [52]. According to other authors, the cause of increased salivary levels of lactoferrin may be the altered permeability of the oral mucosa for plasma components, which is believed to occur in individuals with long-lasting reduced salivation [51]. Dodds et al. believed that high concentrations of lactoferrin in saliva may also be the result of inflammation in the salivary glands [53]. In our research, we did not observe any relationship between the concentrations of lactoferrin in stimulated saliva and the content of *S. mutans* and *L. acidophillus*. Unfortunately, the available literature does not provide information on a relationship between lactoferrin concentration and microbial titer in patients after alloHSCT. Almståhl et al., examining subjects with decreased salivation resulting from various causes, found no correlation between the concentrations of lactoferrin and the growth of *S. mutans* colonies. At the same time, however, they noticed that increased levels of lactoferrin were accompanied by a significant increase in the number of *L. acidophillus* colonies [11]. Other authors observed a significant ability of lactoferrin to inhibit bacteria adhesion to dental hard tissues. As in the case of lysozyme, products with lactoferrin can be used in home and professional oral prophylaxis [54]. However, the dentist should remember that some of these products may contain probiotics. If the patient is immunodeficient (e.g., cGvHD), they should be used with caution. In this case it is recommended to contact a hematologist.

We found a tendency for an increase in the levels of sIgA in mixed saliva, both in resting and stimulated saliva in patients after alloHSCT, compared to healthy subjects. Although the noted differences were not statistically significant, noticeable differences in concentrations of this immunoglobulin between the analyzed groups were observed. Patients after transplantation demonstrated a significantly shifted upper limit of sIgA concentration range towards the maximum values, compared to subjects in the control group. Therefore, it is important to note that some patients after alloHSCT had high and very high sIgA concentrations in mixed saliva. High values of sIgA concentrations in saliva could be the result of, e.g., increased permeability of oral mucosa (one of the symptoms of mucositis). In the late period after alloHSCT, mucositis is a common symptom of oral cGvHD [5,8]. Available publications emphasize that concentrations of sIgA in mixed saliva are not constant values, and the results of studies describing the levels of this immunoglobulin in various clinical situations are often contradictory. The authors point out a high inter-individual variability of sIgA concentrations [20,55], which depends on a number of factors such as age (immunoglobulin concentration increases with age), race, estrogen levels in women [56], or the emotional state of the subjects (a stressful situation is accompanied by decreased levels of sIgA) [57]. Also, physical and mental effort can increase concentrations of sIgA [58]. It was observed that fluctuations in salivary sIgA levels are also associated with some disease conditions, e.g., immunoglobulin concentration increase in lichen planus and chronic recurrent aphthous stomatitis, and a decrease in chronic proliferative candidiasis [59]. Eliasson et al. found a significant increase in the levels of sIgA in saliva coming from labial glands in patients with Sjögren’s syndrome, compared to healthy subjects [56]. On the other hand, Bachanek et al., analyzing the levels of various antibodies in resting saliva of patients treated for haematopoietic hyperplastic diseases, observed significantly lower levels of sIgA in saliva of those patients compared to healthy individuals [60]. Nikifraryam et al., while examining patients with sIgA deficiency in peripheral blood, observed a statistically significant reduction in sIgA concentrations in mixed resting saliva compared to healthy subjects [61]. Conversely, Chrustowicz et al. [55], while examining children undergoing chemotherapy for neoplastic disease, noticed that the therapy did not cause the expected reduction in sIgA concentrations in resting saliva. On the contrary, the authors pointed out that in three out of 34 patients (8.8%) there was a significant, even several-fold increase in the levels of this immunoglobulin, which was similar to the situation observed in our own research. The mentioned authors believed that the relative stability of sIgA levels in saliva during immunosuppressive treatment was perhaps due to a high local secretory potential of the immune system. Analyzing the factors that may affect the level of sIgA in mixed saliva, we found that the concentrations of immunoglobulin increased significantly with a decrease in the rate of secretion of both resting and stimulated saliva. Similar relationships were observed by some authors who studied healthy subjects [62], as well as patients with reduced salivation after head and neck radiotherapy [63] and individuals with Sjögren’s syndrome [56]. Our research shows that the time that elapsed after alloHSCT did not significantly affect concentrations of sIgA, which in some subjects were significantly elevated even after a very long period after the procedure. Completely different observations were made by Imanguli et al., who, when examining saliva of patients after hematopoietic cell allograft, found that the levels of sIgA in mixed saliva were significantly decreased immediately after transplantation, but after six months they returned to the state before the procedure [22]. Also, Steinbrenner et al. noticed that the levels of IgA in mixed resting saliva were the lowest in the period of 3–4 months after transplantation. However, as soon as one year after the procedure was reached, in about 75% of patients they had the same values as those observed in the control group. In contrast, these authors found that peripheral blood IgA levels did not become stable until about 18 months after alloHSCT [64]. In our study, the levels of this immunoglobulin were statistically significantly higher in patients with high titer of *S. mutans*. The obtained results of the study, regarding a possible relationship between severity of caries and sIgA salivary concentrations in patients in the late period after alloHSCT, are unfortunately not referenced in the available literature. On the other hand, the results of the authors’ research analyzing this problem in generally healthy subjects are contradictory. Wu et al., based on a meta-analysis, concluded that high intensity of caries in children is associated with low sIgA concentrations in mixed saliva [65]. However, Zalewska et al. did not observe significant differences in concentrations of sIgA in mixed saliva between generally healthy adults with high and low susceptibility to caries [66]. The authors, who studied subjects with immune system dysfunction caused by factors other than alloHSCT, also drew attention to the problem of caries. The studies conducted by Cole et al. on patients with peripheral blood sIgA deficiencies showed in this group of patients a significantly higher incidence and severity of caries compared to healthy subjects [67]. On the other hand, Nikifarjam et al., while examining a similar group of patients, did not find significant changes in incidence of caries [61].

The current study has its limitations. The presented analysis concerns only some of the components of innate and acquired immunity present in saliva. In order to obtain a more complete picture in patients who are in the late period after alloHSCT, concentrations of other immune proteins (e.g., salivary amylase, cystatins, prolin-rich proteins, mucins, peroxidases, statherin) should also be assessed. In the next publication, we would like to include data on the observed relationships between concentrations of lactoferrin, lysozyme, and sIgA on the one hand and on the other the results of clinical examinations (e.g., occurrence of inflammation of the oral mucosa and tooth decay—DMFT). In addition, the presence or absence of systemic factors influencing saliva properties (mainly cGvHD), should be considered. To confirm our results with more powerful evidence, a large patient population and well-designed trials are necessary in further studies.

## 4. Materials and Methods

### 4.1. Study Population

The characteristics of the study population are presented in Table 5.

Approval for the study was obtained from the Bioethics Committee of the Medical University of Warsaw, Poland, (protocol code KB/54/A/2017). This study was conducted in line with the principles of the Declaration of Helsinki. All patients were referred to the aforementioned dental clinics for oral evaluation as part of standard post-transplantation procedures. Every patient gave consent to participate in the study. The study was conducted in the years 2018–2020. The control group consisted of 30 subjects, selected by age and gender (19–67 years, SD = 38.87 ± 11.76; Me = 36.5), generally healthy, who consented to participate in the study.

We used the following exclusion criteria:-chemotherapy with autologous HSCT,-second alloHSCT,-any acute or chronic condition that would limit the ability of the patient to participate in the study,-smoking,-refusal to give informed consent,-failed saliva collection (severe hyposalivation, gag reflex during collection),-removable dental prostheses.

### 4.2. Assessment of the Rate of Salivation

Mixed resting saliva and mixed stimulated saliva according to accepted standards [68,69] were collected from each patient. Stimulated and resting mixed saliva was taken from all patients before noon (between 9 a.m. and 11 a.m.) and two hours after patients had eaten, washed their teeth, chewed gum or drunk anything. Taking resting saliva consisted in accumulating it in the mouth for two minutes, followed by spitting it into a calibrated test tube of the Corning type. The procedure was repeated three times, which gave the total time of taking saliva equal to 6 min. The total capacity of the obtained saliva was divided by 6, which gave the amount of saliva secreted during one minute (mL/min). Stimulated saliva was obtained by giving a paraffin cube to a patient who was supposed to chew it for six minutes. After one minute of stimulation patients were supposed to swallow the stimulated saliva, and next, to spit parts of the saliva obtained during five minutes into a test tube of the Corning type. The total capacity of the obtained stimulated saliva was converted to the amount obtained during one minute. Immediately after collection, a part of the stimulated saliva was used for testing buffering capacity and for inoculating it on culture media. The obtained resting saliva as well as the remaining amount of stimulated saliva were frozen and stored at a temperature of −30 °C, for further tests [68,69].

### 4.3. Microbiological Assessment of Saliva

For microbiological tests, stimulated saliva was used, which most fully reflects the microbiological status of the oral cavity. The inoculations were made on the following selective culture media:(a)CRT Bacteria from Ivoclar-Vivadent (Lichtenstein) for culturing of *S. mutans*,(b)CRT Bacteria from Ivoclar-Vivadent (Lichtenstein) for culturing of *L. acidophilus*.

Inoculation on these media was performed according to the manufacturer’s recommendations. All saliva samples were incubated for 48 h at 37 °C under aerobic conditions. The numbers of Streptococcus mutans and Lactobacillus acidophilus colonies were read by comparing them with reference cards provided by the manufacturer. The obtained results are presented in the following ranges:(a)<10^5^ CFU/mL of saliva: low colony count; (CFU—colony-forming units),(b)≥10^5^ CFU/mL of saliva: high colony count.

Based on general assumptions, it was presumed that a high risk of caries appears when the number of *S. mutans* and *L. acidophilus* colonies ≥10^5^ CFU/mL.

### 4.4. Determination of Concentration of Lysozyme, Lactoferrin, and sIgA in Mixed Resting and Stimulated Saliva

In centrifuged mixed resting and stimulated saliva, the concentrations of lysozyme, lactoferrin, and sIgA-type antibodies were determined. Two-stage enzyme-linked immunosorbent assay based on the ELISA (enzyme-linked immunosorbent assay) test was used to determine the concentrations of lysozyme and lactoferrin. Human milk lysozyme was used as a reference. The obtained results are presented in μg/mL. The concentrations of sIgA antibodies were tested by one-step enzyme-linked immunosorbent assay based on the ELISA test [70]. The reference was human IgA. The results obtained are presented in µg/mL.

### 4.5. Statistical Analysis

In order to verify the research hypotheses, statistical analyzes were carried out using the IBM SPSS Statistics 25 package. Descriptive statistics, Shapiro–Wilk tests, frequency analyses, Spearman’s ρ rank correlation analyzes, and Mann–Whitney U tests were performed. The classical threshold of α = 0.05 was assumed as the significance level.

## Figures and Tables

**Figure 1 ijms-25-01457-f001:**
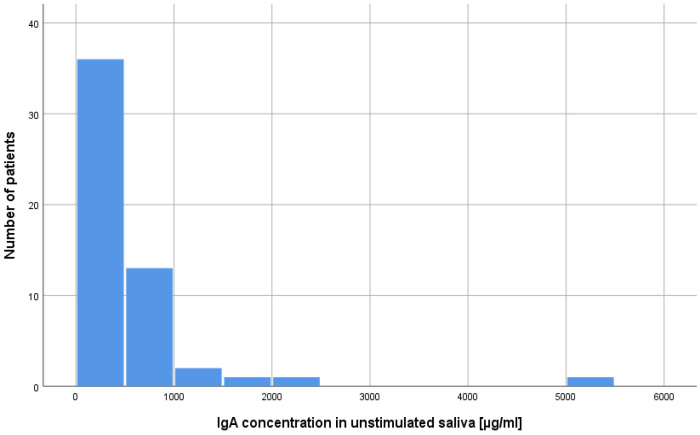
Number of alloHSCT patients with a given concentration of sIgA in unstimulated saliva.

**Figure 2 ijms-25-01457-f002:**
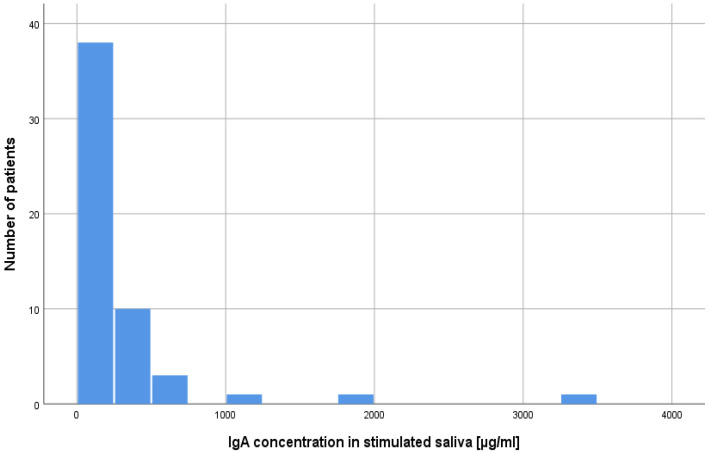
Number of alloHSCT patients with a given concentration of sIgA in stimulated saliva.

**Figure 3 ijms-25-01457-f003:**
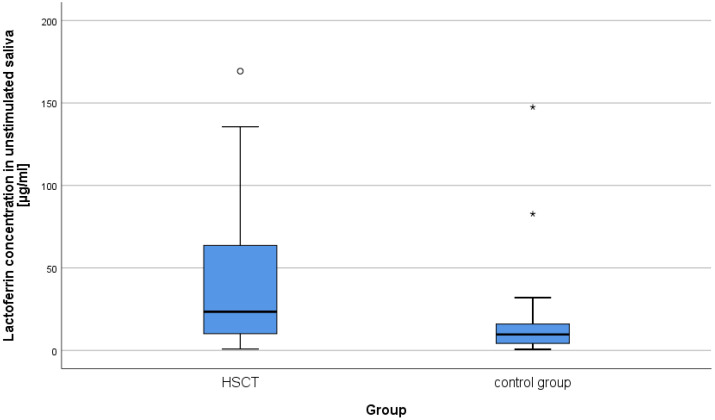
Lactoferrin concentrations in unstimulated saliva in the study and control groups. From the bottom: minimum, lower quartile, median, upper quartile, non-outlier maximum, outlier extreme value. ○—outlier value, *—extreme value.

**Figure 4 ijms-25-01457-f004:**
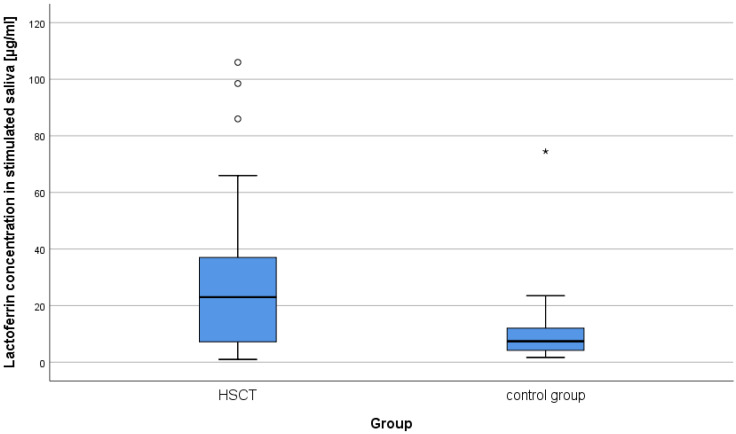
Lactoferrin concentrations in stimulated saliva in the study and control groups. From the bottom: minimum, lower quartile, median, upper quartile, non-outlier maximum, outlier extreme value. ○—outlier value, *—extreme value.

**Table 1 ijms-25-01457-t001:** Concentrations of lysozyme, lactoferrin, sIgA in resting and stimulated saliva in patients after alloHSCT and in the control group.

Selected Components of Saliva	Type of Saliva	alloHSCT (n = 54)			Control Group (n = 30)						
		M	SD	Me	M	SD	Me	U	Z	*p*	r
Lysozyme	unstimulated	13.85	17.03	9.28	13.46	10.24	11.65	757.5	−0.49	0.624	0.05
	stimulated	18.32	26.59	10.05	11.55	8.24	9.03	726.0	−0.78	0.433	0.09
Lactoferrin	unstimulated	40.18	39.83	23.45	17.58	28.99	9.65	417.5	−3.66	<0.001	0.40
	stimulated	27.33	24.70	23	10.69	13.20	7.4	431.5	−3.53	<0.001	0.39
sIgA	unstimulated	512.53	808.54	256.05	286.21	172.36	261.5	735.0	−0.70	0.484	0.08
	stimulated	282.66	521.69	142.49	165.20	94.27	155	800.0	−0.09	0.926	0.01

**Table 2 ijms-25-01457-t002:** Correlation between lysozyme, lactoferrin, sIgA concentrations and the time after alloHSCT.

Selected Components of Saliva	Type of Saliva		Time after alloHSCT
Lysozyme	unstimulated	Spearman’s ρ	−0.12
		significance	−0.388
	stimulated	Spearman’s ρ	−0.02
		significance	0.888
Lactoferrin	unstimulated	Spearman’s ρ	0.01
		significance	0.957
	stimulated	Spearman’s ρ	−0.01
		significance	0.923
sIgA	unstimulated	Spearman’s ρ	0.10
		significance	0.455
	stimulated	Spearman’s ρ	0.09
		significance	0.502

**Table 3 ijms-25-01457-t003:** Correlation between lysozyme, lactoferrin, sIgA concentrations and saliva flow rate in alloHSCT patients and in the control group.

Selected Components of Saliva		Unstimulated Saliva Flow Rate		Stimulated Saliva Flow Rate	
		alloHSCT (n = 54)	Control Group (n = 30)	alloHSCT (n = 54)	Control Group (n = 30)
Lysozyme	Spearman’s ρ	0.05	0.09	−0.04	−0.11
	significance	0.708	0.628	0.771	0.580
Lactoferrin	Spearman’s ρ	−0.41	−0.38	−0.43	−0.21
	significance	0.002	0.036	0.001	0.272
sIgA	Spearman’s ρ	−0.42	−0.41	−0.50	−0.08
	significance	0.001	0.025	<0.001	0.656

**Table 4 ijms-25-01457-t004:** Correlation between lysozyme, lactoferrin, sIgA concentrations in stimulated saliva and *S. mutans* and *L. acidophilus* titers in stimulated saliva.

Selected Components of Saliva		*S. mutans* Count		*L. acidophilus* Count	
		alloHSCT (n = 54)	Control Group (n = 30)	alloHSCT (n = 54)	Control Group (n = 30)
Lysozyme	Spearman’s ρ	0.32	0.22	−0.10	−0.09
	significance	0.017	0.245	0.479	0.649
Lactoferrin	Spearman’s ρ	0.05	0.35	0.18	0.20
	significance	0.729	0.060	0.200	0.291
sIgA	Spearman’s ρ	0.31	0.46	0.01	0.04
	significance	0.021	0.011	0.940	0.848

**Table 5 ijms-25-01457-t005:** Patients characteristics (n = 54).

Characteristics	Number
Patient age, median (range)	39.5 (19–67)
Women	24
Men	30
Time after alloHSCT day 0 (months)	3–96
Patients with cGvHD	36
cGvHD duration (months)	9.98

## Data Availability

Data can be obtained from the corresponding author on request.

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
