# Peer review of "Assessment of the Salivary Concentrations of Selected Immunological Components in Adult Patients in the Late Period after Allogeneic Hematopoietic Stem Cell Transplantation—A Translational Study"

_ijms, 2024, doi:10.3390/ijms25031457_

Round 1
Reviewer 1 Report
Comments and Suggestions for Authors
Abstract
The aim of the study could be reformulated to avoid words as “also”, “additionally” …
Row 21: “(20 corrected to “(2), I presume. Little editing is required.
Introduction
Row 98: The Introduction should not contain any results or conclusion of the present study.
Please add, instead, a hypothesis of the research.
Materials and methods
Row 324: Please be more specific regarding the inclusion/exclusion criteria.
Row 334: There is no mention about measuring the rate of salivation. Instead, the authors mentioned the buffer capacity that I did not find in the other sections.
Results
The text and the figures repeat information.
In my opinion, there are too many figures. If you compress them and change the type of the charts it would be easier for the reader to follow the information. Also, please insert error bars. All figures should have the same format. Reconsider the figures titles.
Row 145: At exactly what periods of time were the biomarkers assayed?
Discussion
Please add a few paragraphs about the mechanisms the authors consider to be involved in the alteration of the assayed biomarkers after alloHSCT.
Also, you could be more specific regarding the justification of the study.
Author Response
Reviewer 1.
Dear reviewer,
Thank you very much for your valid and constructive comments. We hope all the inaccuracies and mistakes has been well corrected.
- The aim of the study has been reformulated.
- Row 21: the mistake has been corrected.
Introduction
This part has been corrected according to suggestions.
Materials and methods
The inclusion/exclusion criteria has been corrected now.
We added more detailed descriptions of methods we used for measuring the rate of saliva. The results regarding buffer capacity were not included in this paper – this information is redundant.
Results
According the suggestion the figures 1-4 have been replaced by the table.
The error bars was insert.
1. The figures have the same format and the titles was reconsider.
- The biomarkers were assayed in the moment each patient was after alloHSCT day 0. We would like to know if there were differences biomarkers concentration in saliva between patients being in different time after the transplantation.
Discussion
This part has been corrected according to the suggestions.
Reviewer 2 Report
Comments and Suggestions for Authors
The authors should present the background knowledge about the impact of Transplantation on saliva parameters. They should describe the pathogenetic mechanism and why they designed this study. They also describe further the research questions. I don’t understand the statements after the aim of this study. They seem to present the results. Do you clarify?
The authors should present the clinical situation. Do you record the DMFT, the presence of active caries, and the plaque index? We shouldn’t interpret and make correlations among the results with those data. The time range after the Transplantation varied from 3-96 months. The authors should add a table with the general characteristics (demographic etc) of the sample. At the same direction, in my opinion they should present the results stratified.
In discussion section, the author mentioned that the one parameter presented higher values, but no statistically significant. They should erase the statement higher.
Comments on the Quality of English LanguageMinor editing of English language required
Author Response
Reviewer 2
Dear reviewer,
Thank you very much for your valid and constructive comments. We hope all the inaccuracies and mistakes has been well corrected.
We put the background knowledge about the impact of allogeneic stem cell transplantation on saliva parameters in the text. We hope, we better explained the aim and design of the study now. We also add a table with the general characteristics of the sample.
The clinical situation of every patient was thoroughly recorded. But we are going to publish them in another paper.
Reviewer 3 Report
Comments and Suggestions for Authors
Manuscript of considerable interest for the dental sector, requires a major revision before being evaluated for future publication.
Title: being a translational study, I would also include the term in the title.
Abstract: the results obtained are missing, highlight them more
Keywords: few, add others that are registered on MeSH.
Introduction, all the causes that can induce carious lesions (Scribante et al) and all the remineralization systems that can be modulated based on the salivary components, such as fluoride and biomimetic hydroxyapatite, compared by the same research group are missing.
Very confusing results, reorganize them in order to make them usable for the reader.
Discussions: add home use to increase the % of lactoferrin through toothpastes or food supplements. (Butera et al.)
Conclusions: modify them according to the changes in the text.
Bibliography: add required references
Author Response
Reviewer 3
Dear reviewer,
Thank you very much for your valid and constructive comments and for your nice opinion. We hope all the inaccuracies and mistakes has been well corrected.
Title: we put term “translational” in the title
Abstract: We have highlight the obtained results.
Keywords: We added some more keywords.
Introduction: It has been corrected according to the suggestions.
The results was reorganised
Discussions: according to the suggestions.
Conclusions: It has been corrected according to the changes in the text.
Bibliography: Required references are added.
Round 2
Reviewer 1 Report
Comments and Suggestions for Authors
Dear authors,
Thank you for accepting my recommendations and acting accordingly. All aspects, I considered should be modified, have been taken cared of.
Reviewer 2 Report
Comments and Suggestions for Authors
The reviewers responded the points had arisen. The quality of the manuscript had improved and I have no further comments.
Comments on the Quality of English LanguageEnglish language fine. No issues detected
Reviewer 3 Report
Comments and Suggestions for Authors
The manuscript has been correctly revised according to the comments provided in the first revision round, it can be published